# Group-based local adaptive deep multiple kernel learning with lp norm

**Shengbing Ren** *, **Fa Liu**, **Weijia Zhou, Xian Feng, Chaudry Naeem Siddique**

School of Computer Science and Engineering, Central South University, Changsha, China

☯ These authors contributed equally to this work.

* rsb@csu.edu.cn

## Abstract

The deep multiple kernel Learning (DMKL) method has attracted wide attention due to its better classification performance than shallow multiple kernel learning. However, the existing DMKL methods are hard to find suitable global model parameters to improve classification accuracy in numerous datasets and do not take into account inter-class correlation and intra-class diversity. In this paper, we present a group-based local adaptive deep multiple kernel learning (GLDMKL) method with lp norm. Our GLDMKL method can divide samples into multiple groups according to the multiple kernel k-means clustering algorithm. The learning process in each well-grouped local space is exactly adaptive deep multiple kernel learning. And our structure is adaptive, so there is no fixed number of layers. The learning model in each group is trained independently, so the number of layers of the learning model maybe different. In each local space, adapting the model by optimizing the SVM model parameter $\alpha$ and the local kernel weight $\beta$ in turn and changing the proportion of the base kernel of the combined kernel in each layer by the local kernel weight, and the local kernel weight is constrained by the lp norm to avoid the sparsity of basic kernel. The hyperparameters of the kernel are optimized by the grid search method. Experiments on UCI and Caltech 256 datasets demonstrate that the proposed method is more accurate in classification accuracy than other deep multiple kernel learning methods, especially for datasets with relatively complex data.

## Introduction

Because different kernels have different characteristics and different parameter settings, the performance of the kernels will be very different on different datasets. And there is no good way to construct or choose a suitable kernel. To solve the problem of these kernels, multiple kernel learning (MKL) method using a combination of kernels has been proposed [1–7], which makes full use of the characteristics of various kernels, and adapts better to different datasets. But in many cases, these combinations of multiple kernel learning don't change the kernel structure. So how to choose the right basic kernel to combine into a composite kernel is still a major issue.

**Data Availability Statement:** UCI and Caltech 256 datasets are public respositories. https://archive.ics.uci.edu/ http://www.vision.caltech.edu/Image_Datasets/Caltech256/.

**Funding:** The author(s) received no specific funding for this work.

**Competing interests:** The authors have declared that no competing interests exist.

Multiple kernel learning is combined with deep learning [8, 9] to improve learning performance. The deep learning method transforms the input data through multiple nonlinear processing layers to construct a new feature [10]. These methods have successfully made significant progress in image classification [11]. There are many related studies on deep multiple kernel learning(DMKL). Deep multiple kernel learning aims to learn the "deep" kernel machine [12] by exploring a combination of multiple kernels in a multilayer structure. Through multilevel mapping, the proposed multi-layer multiple kernel learning (MLMKL) framework is more adaptable to a wide range of datasets than the MKL framework to find the optimal kernel more efficiently. In [13], the combined kernel is formed through the base kernel in each layer and optimizing the estimate of forgetting errors in support vector machines. The structure is directly the mutual weighted iteration of different combined kernels that results in too many weight parameters, and it is too cumbersome to optimize parameters. In [14], a backpropagation MLMKL framework is proposed, which uses deep learning to iteratively learn the optimal combination of kernels. Three deep kernel learning models for breast cancer classification problems have been proposed in [11] to avoid overfitting risks in deep learning. However, These model structures have a fixed number of layers and a lack of flexibility. Moreover, the model learning methods are global, without high generalization ability under certain conditions.

Because classical DMKL models have a fixed number of layers, such models can not adapt to a wide range of datasets. As a result, the classification performance is not the best, and wasting of computing power, needing more data, and so on. Therefore, in [15], we propose an adaptive deep multiple kernel learning framework, which solves the problem of the model with a fixed number of layers. Increasing the number of layers according to the actual datasets, and the cutoff condition for increasing the number of layers is that the highest classification accuracy is continuously unchanged for several layers. However, the adaptive deep multiple kernel learning requires too many layers to achieve the highest accuracy which wastes time. And the classification performance is greatly affected by the type of kernels, resulting in poor model stability.

Classical DMKL is limited to learning the global combination of the entire input space. Due to the diversity and correlation between samples, suitable kernels may vary from one local space to another. When the samples in a category exhibit high variation as well as correlation with the samples in other categories, they are difficult to cope with such complicated data and suffer degraded performance. So we introduce the local learning method [16] to solve the problem.

The local learning method can take into account the inter-class correlation and intra-class diversity. Moreover, we can divide the datasets into several groups by the clustering algorithm to facilitate the classification and statistical analysis of subsequent models. Moreover, it can be regarded as only one group for the datasets with very simple samples so that DMKL can be carried out directly. It can be seen that the local learning method is very adaptable to a wide range of datasets and can reduce the complexity of the model and save training time. Therefore, it is very feasible to apply the local learning method to DMKL to form local deep multiple kernel learning based on grouping, which can improve the generalization performance of the DMKL model and is higher than classical DMKL in classification accuracy. Another benefit is to save computing power and not need too much data.

In group-based local deep multiple kernel learning, samples with similar are clustered into a group so that the intra-class diversity can be represented by a set of groups. In addition, inter-class correlation can be represented by the correlation among the different groups. So group-based local deep multiple kernel learning can be adapted to a wide range of datasets to increase the flexibility of the model and save computing power. Another advantage of group-

based local deep multiple kernel learning is that multiple classifiers can be trained separately, and the classifier model layers in each group may be different. In other words, each group is performed separately so that saving training and prediction time. Only need to know which group the new test sample falls into, you can test in the local model of the corresponding group, and calculate the classification accuracy, which helps to adapt to a variety of samples and highlights the flexibility of the model.

Because the sparse constraint can lose useful kernels during MKL optimization [17], we utilize the lp norm [18] constraint on the kernels and get non-sparse results to avoid losing useful kernels. Therefore, the lp norm will be used for weight constraints so that the weight of useful kernels will be increased. As a result, useful kernels will not lead to the loss. On the contrary, the weight will be reset to zero for the useless or even counterproductive kernels. In this way, the kernel sparsity is adjusted in the multiple kernel combination in each layer can also improve the classification performance.

To solve the above problems, this paper proposes a group-based local adaptive deep multiple kernel learning (GLDMKL) method with lp norm. Unlike classical DMKL, our GLDMKL model is based on local learning and adaptive. So samples are clustered using the multiple kernel k-means clustering algorithm so that the similar samples are in the same group. For those samples after they are divided into multiple groups, the DMKL process is performed in the respective local spaces. The number of layers in each group may be inconsistent, which highlights the flexibility of the model and saves training and testing time. In each group at each layer, we perform a MKL process by weighting multiple kernels with different types and parameters to form a combined kernel. Moreover, determining the proportion of the basic kernel in each combined kernel according to the local kernel weight. In local adaptive deep multiple kernel learning, the output value of the combined kernel in the previous layer is used as an input to the combined kernel in the next layer. However, the actual input of the local adaptive deep multiple kernel learning is still a sample. We can study the deep kernel machine through the above methods and can stop increasing the number of layers as long as the highest classification accuracy is continuously unchanged for several layers. Moreover, our model needs to set an initial value for each candidate kernel hyperparameter, and we adjust it by grid search method [19] to avoid the trouble of manually selecting kernel hyperparameters before the learning process. Also, this learning method is to constrain the weight with the lp norm and to control the sparsity of kernel to avoid losing useful kernel. For the useless kernel, the weight can be reset to zero. Thus, multiple kernel combinations of non-sparse kernels in each layer can improve generalization capabilities.

The main contributions of this paper are summarized as follows: (1) A group-based local adaptive deep multiple kernel learning architecture is proposed. The GLDMKL architecture consists of two parts: multiple kernel k-means clustering and local adaptive deep multiple kernel learning. Furthermore, the number of layers grows with the learning process. In each group, the learning process is carried out independently, and the number of layers of the learning model may be different. Our model is more adaptable to data of different dimensions and sizes. (2) A GLDMKL model learning algorithm to adapt the architecture is designed. Our model learning algorithm utilize deep kernel learning to build a local deep multiple kernel learning model layer by layer. And the SVM model parameters and local kernel weights corresponding groups are optimized in turn to fit the model. The hyperparameters of basic kernels are adjusted by the grid search method. Also, stopping the growth of the model layers is determined by the highest classification accuracy invariant in continuous layers. (3) The weight constraint with the lp norm is proposed. For the sake of controlling the sparsity of the kernel and avoiding the loss of useful basic kernels, the weight of useful basic kernels will be increased. (4) Experiments on UCI dataset and Caltech 256 dataset show that our GLDMKL

approach has the power to handle complex data. And compared with classical DMKL methods, our GLDMKL method has higher classification accuracy and higher generalization.

The rest of this paper is organized as follows: Section Related works provides a brief overview of the relevant background. Then a group-based local adaptive deep multiple kernel learning method with lp norm is described in Section Our approach. Section Experiments describes the experimental part. Section Validation illustrates the validation of the model. Section Conclusion provides a summary of the paper and future work.

## Related works

### Deep multiple kernel learning

Deep multiple kernel learning(DMKL) [12–15, 20–24] is a hot research topic inspired by deep learning in recent years. This method explores the combination of multiple kernels in a multi-layer architecture and achieves success on various datasets. Therefore, DMKL can be used in many real-world situations.

In [12], Zhuang et al. propose a two-layer multiple kernel learning (2LMKL) method and two efficient algorithms for classification tasks. It aims to learn "deep" kernel machines by exploring a combination of multiple kernels in a multi-layer structure. With multi-layer mapping, the proposed 2LMKL framework provides greater flexibility than conventional MKL for finding the best combined kernels faster. Zhuang et al. also show that the number of basic kernels has a certain effect on the classification performance, and it is realized by iteratively updating the parameters of the basic kernel. However, there are only two layers of structure, which cannot adapt to the requirements of a wide range of datasets and the model is global.

In [13], a combined kernel is formed by the basic kernel at each layer and then optimizing over an estimate of the support vector machine leave-one-out error. There require only a few basic kernels to continuously improve performance at each layer. Its structure is directly weighted iteration through different kinds of combined kernels, resulting in too many weighting parameters. It is too cumbersome to optimize parameters and the number of model layers is fixed. Moreover, the model is also global, without taking into account intra-class diversity and inter-class correlation.

In [14], a new backpropagation MLMKL framework is described, which optimizes the network through an adaptive backpropagation algorithm. Rebai et al. use the gradient ascent method instead of the dual objective function. The deep architecture has a fixed number of layers and cannot adapt to a wide range of datasets. And it's also a global model, without taking into account intra-class diversity and inter-class correlation.

In [15], we propose an adaptive deep multiple kernel learning (SA-DMKL) method. It can optimize the model parameters of each kernel with the grid search method. And each basic kernel is evaluated using a generalization boundary based on Rademacher chaotic complexity and those that exceed the generalization boundary are removed. The output regression value of the SVM classifier constituted by other kernels is used to construct the new feature space. The dimension of the new feature space is the number of the remaining kernels, thus forming the new sample data features as the input of the kernels in the next layer. And the SVM classifier is used to train each candidate kernel. At the same time, in each layer, the SVM classifier based on the kernel is used to classify test data and obtain classification accuracy. The growth of the layers is terminated by the highest classification accuracy unchanging in successively several layers. But the model is also global, without taking into account intra-class diversity and inter-class correlation.

DMKL has a good effect on generalization ability when candidate kernels and parameters are adjusted to a very appropriate level. That effect, however, is hard to achieve. Many

hyperparameters need to be set and are difficult to adjust. At the same time, the existing DMKL architecture is relatively simple. The combined kernel in each layer consists of a set of the same basic kernels. And the output of the combined kernel in the previous layer is the input of all the basic kernels in the next layer. Also, the number of layers is fixed. The proper selection of kernel and model structure with a fixed number of layers lead to insufficient adaptability to the sample datasets, which affects the performance of the model.

So, we propose an adaptive DMKL architecture [15] to solve the problem of a fixed number of layers. The growth of layers can be limited by setting a cutoff condition, and model layers can change with different training datasets.

However, there is also a problem that these learning methods adopt a uniform similarity measure over the whole input space. When the samples of a category exhibit high variation as well as correlation with other categories, they are difficult to cope with such complex data.

## Group-based local learning

Local learning [16, 25–28] is to divide the whole problem into several small problems, then learning separately. Local learning only needs to find the local optimum, which is more convenient and more efficient than global learning. We apply group-based local learning to DMKL instead of global learning. The following is a description of group-based local learning and global learning.

A comparison of group-based local learning and global learning architecture is shown in Fig 1.

The original sample dataset is represented as *Data*, the number of basic kernels are $m$ and $\{k_1, k_2, \ldots, k_m\}$ are basic kernels. And each kernel has a weight. The difference between the two is:

1. Group-based local learning divides the sample dataset into multiple groups and performs MKL in each group, where the number of groups is $g$, $\{G_1, G_2, \ldots, G_g\}$ represent g groups which have several samples and the total number of weights is $g^* m$;

2. Global learning is classical MKL for sample dataset, where the total number of weights is $m$.

Here are the benefits of group-based local learning:

How to choose the proper kernel is very difficult; it involves the selection of the hyperparameters of basic kernels. There are also a variety of basic kernels, and the weight setting of basic kernels. With so many parameters, it is impossible to select the best combination of parameters quickly.

Group-based local learning is also based on multiple kernel learning, but there is no strict need to select the most appropriate kernels for multiple kernel learning. And local learning is carried out by clustering to ease the computational pressure of choosing the right kernels. Another advantage is taking into account inter-class correlation and intra-class diversity and having the ability to deal with complex data. So group-based local learning is a very desirable method.

## Lp norm

Norm [29] is a reinforced notion of distance, which by definition adds a scalar multiplication algorithm to distance. Sometimes we can think of the norm as a distance for the sake of understanding.

In mathematics, the norm includes the vector norm and the matrix norm. The vector norm represents the size of the vector in the vector space, and the matrix norm represents the size of the change caused by the matrix. A non-strict interpretation is that corresponding vector

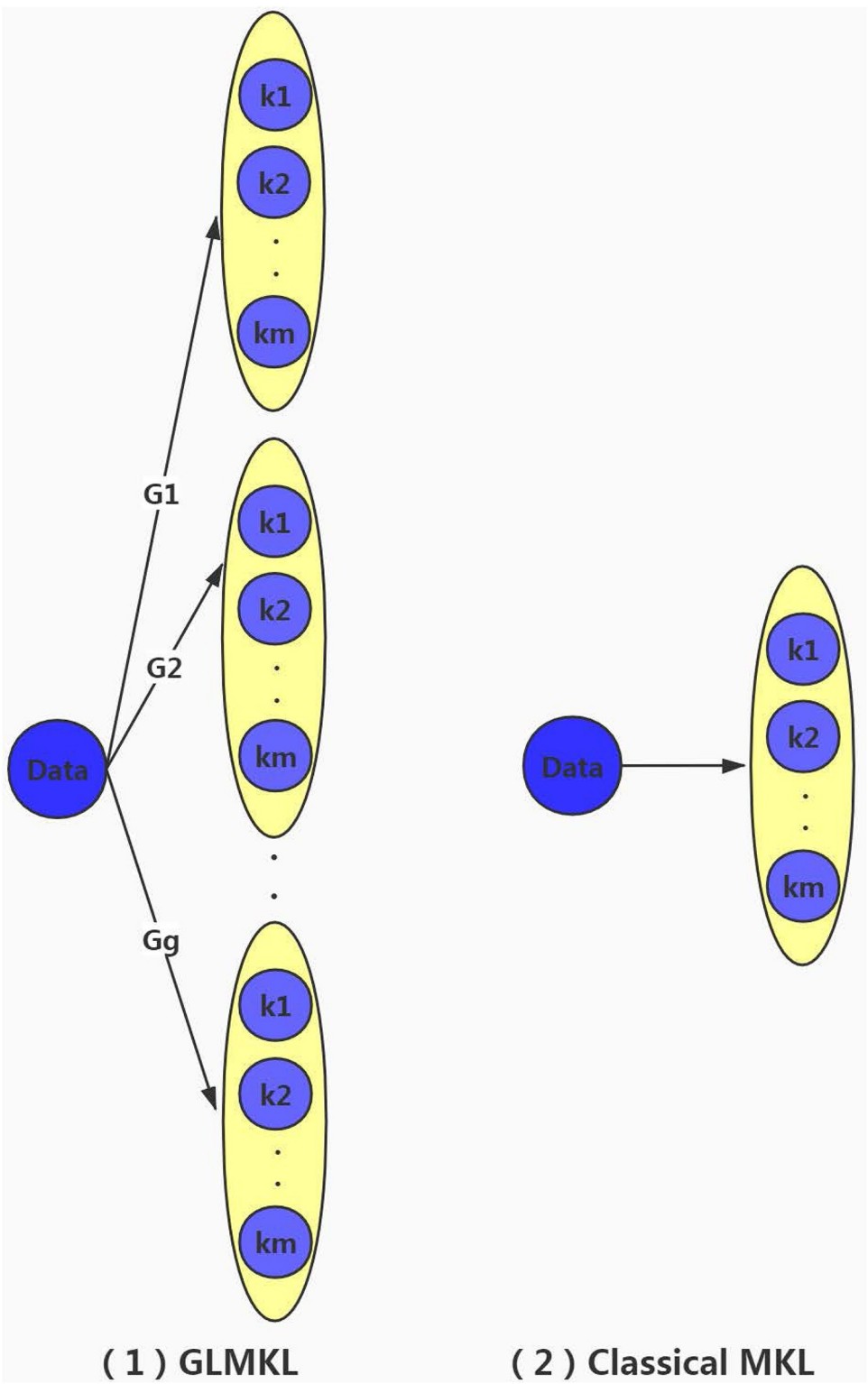

**Fig 1. Comparison of two multiple kernel learning methods.**

norms, vectors in vector space are of magnitude. How to measure this size is measured by the norm. Different norms can measure this size, just like both meters and feet can be used to measure distances. We know that by computing AX = BAX = B, vector X can be changed to B, and the matrix norm is used to measure the magnitude of this change.

And the lp norm [30–32]is defined as Eq (1), $p$ takes the range of $[0, +\infty)$.

$$lp = \sqrt[p]{\sum_{1}^{n} x_i^p}, \quad x = \{x_1, \ldots, x_n\} \tag{1}$$

When $p = 0$, the lp norm is namely the $l_0$ norm. The $l_0$ norm is not a true norm, which is mainly used to measure the number of non-zero elements in the vector.

The $l_1$ norm has many names, such as Manhattan distance, the smallest absolute error, and so on. Use the $l_1$ norm to measure the difference between two vectors, such as the Sum of Absolute Difference.

The $l_2$ norm is the most common and commonly used. The most metric distance we use is the Euclidean distance, which is the $l_2$ norm. And $l_2$ can also measure the difference between vectors, such as the Sum of Squared Difference.

When $p = \infty$, the lp norm is the $l_\infty$ norm, it is mainly used to measure the maximum value of the vector element.

In conclusion, the lp norm is a commonly used regularization term, where the $l_2$ norm $\|\omega\|_2$ tends to balance the components of $\omega$ as much as possible, i.e. the number of non-zero components is as dense as possible. The $l_0$ norm $\|\omega\|_0$ and $l_1$ norm $\|\omega\|_1$ tend to be as sparse as possible for $\omega$, i.e. the number of non-zero components is as small as possible.

## The sparsity of the kernel

Sparsity regularized multiple kernel learning has been proposed [33–36]. Dong et al. propose a simple multiple kernel learning framework for complicated data modeling, where randomized multi-scale Gaussian kernels are employed as base kernels and a $l_1$-norm regularizer is integrated as a sparsity constraint for the solution.

Sparsity refers to the proportion of the number of non-zero elements. If there are more non-zero elements for zero elements, it is dense; If there are fewer non-zero elements for zero elements, it is sparse.

The concept of the sparsity of the kernel is introduced, and the sparsity of the kernel refers to the number of kernels used. Sometimes, because of the sparse constraint, the useful kernel may be lost in multiple kernel learning optimization. To improve the sparsity of the useful kernel so that the lp norm will be adopted. And the sparse constraint can be implemented by changing the weight. Therefore, lp norm will be used for weight constraint, so that the weight of the useful kernel will be increased, without causing loss. Conversely, for the useless or even counteracting kernel, its weight is reset to zero.

## Our approach

### Architecture

In local deep multiple kernel learning, multiple kernels are combined and the advantages of each kernel are used in each local space. The MKL architecture diagram is shown in Fig 2.

The number of basic kernels is $m$ and $\{k_1, k_2, \ldots, k_m\}$ are basic kernels. And each kernel has a local kernel weight; they are respectively $\{\beta_1, \beta_2, \ldots, \beta_m\}$. $K$ is the combined kernel.Our model is based on several MKL components. And MKL is the core element of our model.

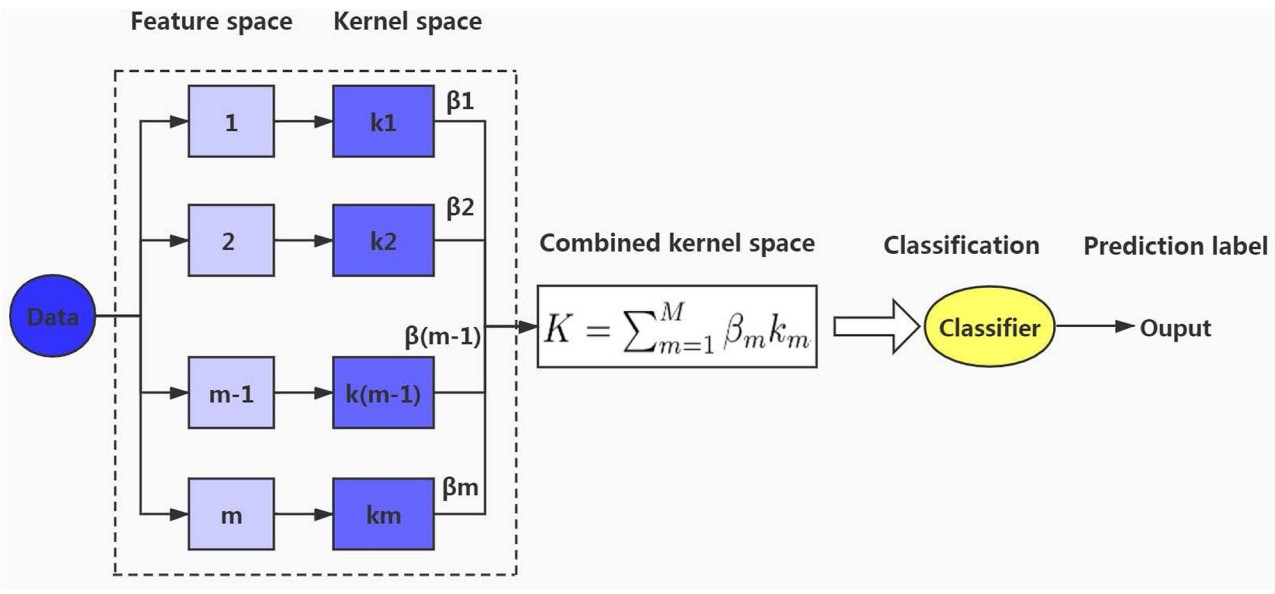

**Fig 2. Multiple kernel learning.**

Since samples consist of multiple features, we propose a multiple kernel k-means clustering method to make the clustering results more reliable. Samples are divided into $g$ groups by clustering, and they are closer together in each group. According to the localization idea, we cluster samples into groups before the first layer network and then optimize the local kernel weight in each group. The purpose of grouping is to make full use of the feature similarity and diversity among samples. So making the learning method more applicable to a wide range of sample datasets. Therefore, we use a group-based local deep multiple kernel learning method.

In our GLDMKL model, the output of the previous layer is used as the input of the next layer to construct a DMKL network. The local space in each group is performed a DMKL process. And our local deep multiple kernel learning is an adaptive structure, which is based on the actual situation. In the local space of each group, the number of layers in each learning process may be different. Moreover, layers' growth is stopped when the highest classification accuracy of several successive layers is unchanged. This prevents the model from constantly growing, wasting time and storage space, and effectively reduces the complexity of the model.

Our model needs to set an initial value for each candidate kernel hyperparameter and adjusts it with a grid search method to avoid manually selecting kernel hyperparameters before the learning process. The kernel's weight parameters are initialized by the gate function to randomly select a relatively small number. The weight parameters are adjusted by the lp norm constraint to obtain non-sparse results to avoid losing useful kernels. Weights are used to adjust the proportion of the basic kernels and we reset the weights to zero for useless kernels. If the weight parameter settings are not appropriate, our model learning algorithm can adjust the combined kernel structure of the next layer by changing the kernel weight.

Our GLDMKL architecture is shown in Fig 3. Before the learning process, the multiple kernel k-means clustering algorithm is used to cluster the training data $Data$, the number of groups is set to $g$, and the training data is divided into $\{D_1, D_2, \ldots, D_g\}$, The number of final layers $L$ in each group is respectively $\{L_1, L_2, \ldots, L_n\}$. The number of basic kernels is $m$, and $\{k_1, k_2, \ldots, k_m\}$ are basic kernels. And each kernel has a local kernel weight and they are respectively $\{\beta_1, \beta_2, \ldots, \beta_m\}$ which are shown in detail in Fig 2. $K_g(L_n)$ is represented as the combined

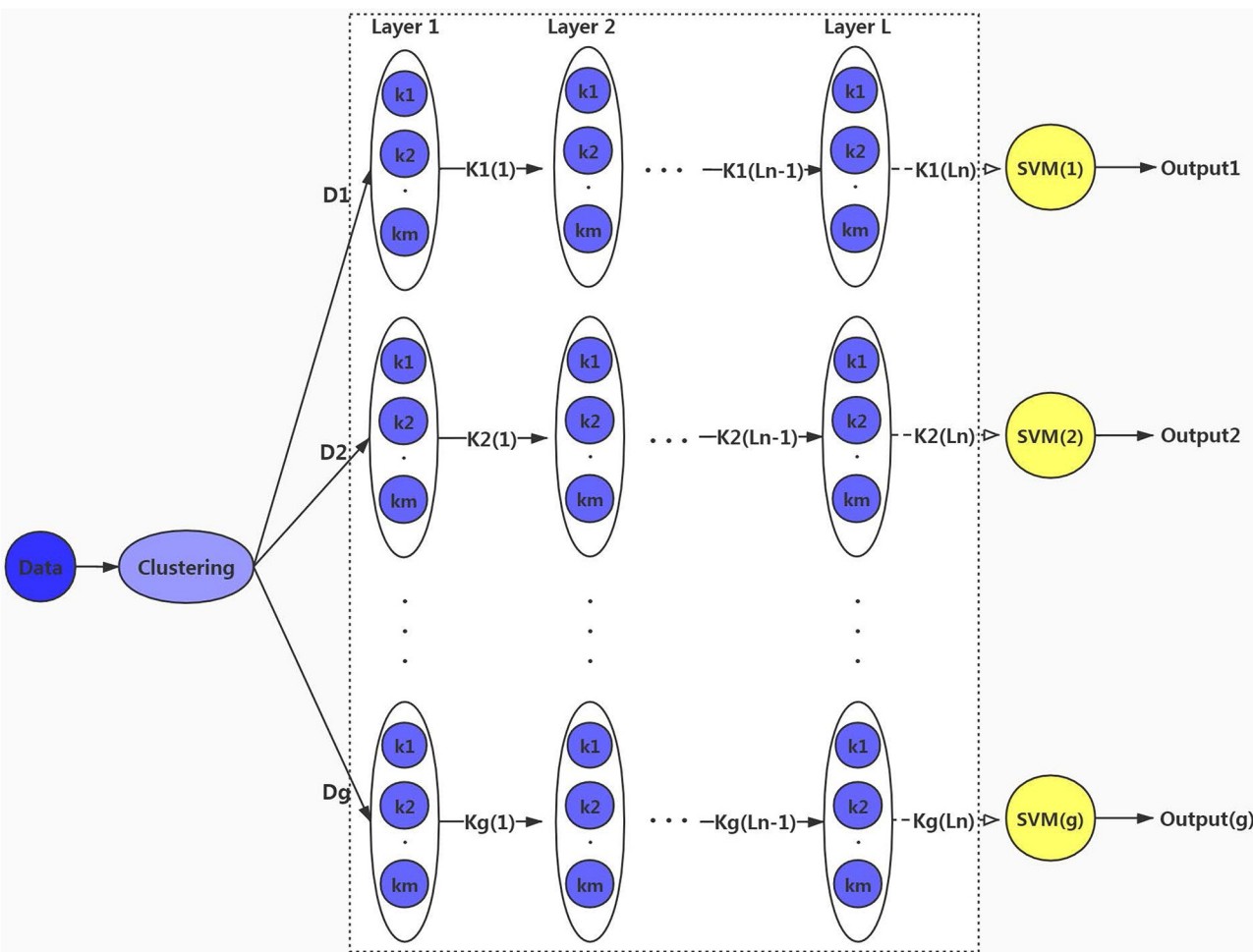

**Fig 3. GLDMKL architecture.**

kernel in layer $L_n$ in group $g$. $g$ groups correspond to $g$ SVM classifiers, and there will also be $g$ output values.

In our GLDMKL architecture, the closer samples were assigned to the same group. In the local space for each group, a SVM classifier that has a multi-layer structure is separately trained. The combined kernel in each layer is composed of a weighted sum of several basic kernels. In local space, the weighted sum of basic kernels is conducted in the previous layer and the output value is used in the previous layer as an input of the combined kernel in the next layer. The input of the actual learning process is still a sample. The multi-layer MKL forms a SVM and samples are classified at the same time. As the model layer continues to grow, the SVM classifier is being updated. The cutoff condition of model layer growth is implemented by the highest classification accuracy unchanged in lasting several layers, thus forming the final SVM classifier model. During the test, we use the clustering algorithm to determine which group samples belong to, then the classification prediction is made in the trained classifier model in the corresponding group, and calculating the classification accuracy in each layer.

Because the classical DMKL model has a fixed number of layers and has obvious limitations, it doesn't take into account intra-class diversity and inter-class correlation and the

model flexibility is poor. The adaptive change of layers not only can increase model flexibility but also can improve the classification accuracy. At the same time, different model layers are performed according to different datasets, which is convenient for reducing model training and prediction time. The key to the adaptive layer is the cut-off condition. As long as the highest classification accuracy remains unchanged in several layers, model layers stop growing. Therefore, we adopt a group-based local adaptive deep multiple kernel learning method.

Because sparse constraints can lose useful kernels, we use the lp norm constraints on kernels and obtain non-sparse results to avoid losing useful kernels. Therefore, we propose a group-based local adaptive deep multiple kernel learning architecture with the lp norm to solve these problems.

## Clustering

In our GLDMKL method, there is a clustering process before training the SVM classifier. We need to design an effective clustering algorithm for our GLDMKL. Since the training samples are represented by multiple features, the traditional clustering algorithms are unable to cluster accurately for a wide variety of samples. In this section, we design a multiple kernel k-means clustering algorithm. Fig 4 shows a complete flow chart about multiple kernel k-means clustering.

We weight the sum of $m$ RBF kernels to form a combined kernel which becomes an element of the sample distance matrix. And the k-means clustering algorithm is used to cluster the input samples. The weight of the RBF kernel is obtained by the centered KTA(CKTA) [37]. CKTA is a novel kernel alignment that performs well in evaluating kernels. The center of the

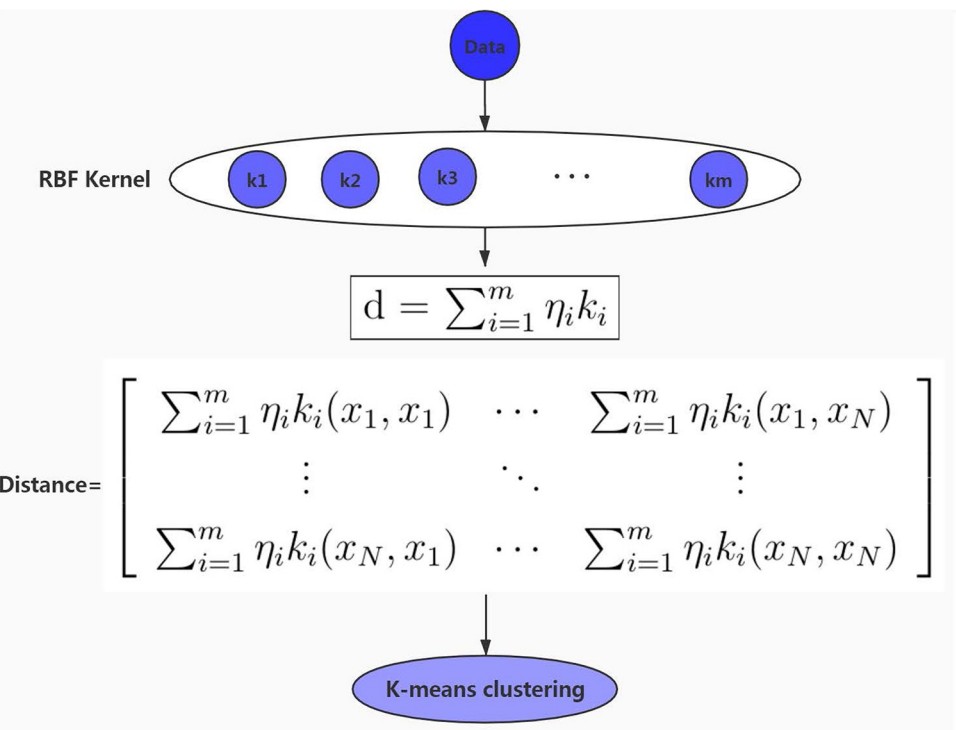

**Fig 4. Multiple kernel k-means clustering.**

 

RBF kernel matrix is Eq (2).

$$k_{ci} = [I - \frac{ee^T}{N}]k_i[I - \frac{ee^T}{N}] \tag{2}$$

Where $e \in \mathbb{R}^{n \times 1}$ denote the vector with all entries equal to one, $I$ denotes the identity matrix, $N$ represents the total number of samples, $k_i$ is the $i$-th RBF kernel matrix, $k_{ci}$ is the center of the $i$-th RBF kernel matrix.

Then we calculate the weight of the RBF kernel matrix as Eq (3).

$$\eta_i = \frac{F(k_{ci}, y)}{\sum_{j=1}^{m} F(k_{cj}, y)} \tag{3}$$

where

$$F(k_{ci}, y) = \frac{\langle k_{ci}, yy^T \rangle_F}{\sqrt{\langle k_{ci}, k_{ci} \rangle_F \langle yy^T, yy^T \rangle_F}} \tag{4}$$

Where $\langle \cdot, \cdot \rangle_F$ denotes the Frobenius product, and $\langle A, B \rangle_F = Tr[A^T B]$. And $y$ is the vector of $\{-1, +1\}$ labels for the sample.

The $m$ RBF kernel matrices are combined into a combined matrix, as shown by Eq (5).

$$K = \sum_{i=1}^{m} \eta_i k_i \tag{5}$$

The combination matrix $K$ is taken as the distance matrix between samples, and then k-means clustering is implemented on $K$. The clustering error can be calculated by the Eq (6).

$$E(C_1, \ldots, C_G) = \sum_{n=1}^{N} \sum_{g=1}^{G} I(\mathbf{x}_n \in C_g) \| \mathbf{x}_n - C_g \|^2 \tag{6}$$

where

$$\| \mathbf{x}_n - C_g \|^2 = k_{nn} - \frac{2 \sum_{l=1}^{N} I(\mathbf{x}_n \in C_g) k_{ln}}{\sum_{l=1}^{N} I(\mathbf{x}_n \in C_g)} + \frac{\sum_{i=1}^{N} \sum_{j=1}^{N} I(\mathbf{x}_i \in C_g) I(\mathbf{x}_j \in C_g) k_{ij}}{\sum_{i=1}^{N} \sum_{j=1}^{N} I(\mathbf{x}_i \in C_g) I(\mathbf{x}_j \in C_g)} \tag{7}$$

Where $G$ is the number of groups, $C_g$ denotes the clustering center of group $g$ and $k_{ij}$ is the distance matrix between sample $\mathbf{x}_i$ and $\mathbf{x}_j$.

Finally, the clustering result for sample $\mathbf{x}_i$ is calculated by the Eq (8).

$$C(\mathbf{x}_i) = \arg\min_{g} (\| \mathbf{x}_i - C_g \|^2) \tag{8}$$

In conclusion, the details of a multiple kernel k-means clustering algorithm are as follows:

1) Starting with CKTA for the kernel alignment, then calculating the kernel weight $\eta_i$. 2) $m$ kernels are RBF kernels with different parameters. 3) The weighted multiple kernel combination between sample pairs is taken as the elements of the distance matrix; then the conventional k-means clustering algorithm is used according to the distance difference between the sample pairs. 4) According to the initiation condition of inputting $G$ groups, it is equivalent to $G$ clustering centers. Then running multiple cycles, and finally clustering into $G$ groups. So

 

samples are closer in each group. 5) The well-grouped samples are used as input for the following learning process.

## GLDMKL

Our model is inspired by soft-clustering-based local multiple kernel learning [38], and our model deals with multiple layer learning problems. To make it easier to understand, so we briefly describe the process of training and testing samples in layer $l$ in GLDMKL, as shown in Fig 5.

**Definition 1** Suppose we have $N$ training samples, and the training dataset is represented by $D = \{\boldsymbol{x}_i, y_i\}_{i=1}^N$, where $\mathbf{x}_i$ represents the $i$-th training samples, $y_i \in \{-1, +1\}$ is the label of the $i$-th training sample. And $\mathbf{x}_i$ can be thought of as a vector consisting of $d$ features. In our GLMDKL, there is a clustering process before classification. So the discriminant function $f^{(l)}$ in layer $l$ is defined as Eq (9).

$$f^{(l)}(\boldsymbol{x}) = \sum_{m=1}^M \beta_{c(\boldsymbol{x}),m}^{(l)} \langle \omega_m, \phi_m(\boldsymbol{x}) \rangle + b \tag{9}$$

Where $\omega_m$ and $b$ is the model parameter, $\beta_{c(\boldsymbol{x}),m}^{(l)}$ represents the weight of the $m$-th kernel of the group $c(\mathbf{x})$ where sample $\mathbf{x}$ is located in layer $l$.

**Definition 2** By modifying the original SVM classifier using this new discriminant function $f^{(l)}$, the training process can be implemented by solving the following optimization problem Eq (10).

$$\begin{aligned}
\min_{\omega_m, b, \xi_i, \beta} \quad & \frac{1}{2} \sum_m^M \| \omega_m \|^2 + C \sum_{i=1}^N \xi_i \\
\text{s.t.} \quad & y_i \left( \sum_{m=1}^M \beta_{c(\boldsymbol{x}_i),m}^{(l)} \langle \omega_m, \phi_m(\boldsymbol{x}_i) \rangle + b \right) \geq 1 - \xi_i \quad \forall i, \\
& \xi_i \geq 0 \quad \forall i, \\
& \sum_m \left( \beta_{c(\boldsymbol{x}_i),m}^{(l)} \right)^P = 1 \quad \beta_{c(\boldsymbol{x}_i),m}^{(l)} \geq 0 \quad \forall i, m
\end{aligned} \tag{10}$$

Where $C$ is the penalty factor, $\xi_i$ is the slack variable, and p represents lp norm to constrain weight.

**Definition 3** Inspired by the original SVM, the Lagrangian multiplier method is used to solve the dual problem of the Eq (10). We first fix the kernel weight $\beta$ and minimize the problem Eq (10). The Lagrangian objective function is represented as a Eq (11).

$$\begin{aligned}
L = \quad & \frac{1}{2} \sum_m^M \| \omega_m \|^2 + \sum_{i=1}^N (C - \alpha_i - \gamma_i)\xi_i + \\
& \sum_{i=1}^N \alpha_i - \sum_{i=1}^N \alpha_i y_i \left( \sum_{m=1}^M \beta_{c(\boldsymbol{x}_i),m}^{(l)} \langle \omega_m, \phi_m(\boldsymbol{x}_i) \rangle + b \right)
\end{aligned} \tag{11}$$

Where $\alpha_i \geq 0$ and $\gamma_i \geq 0$ are Lagrangian multipliers.

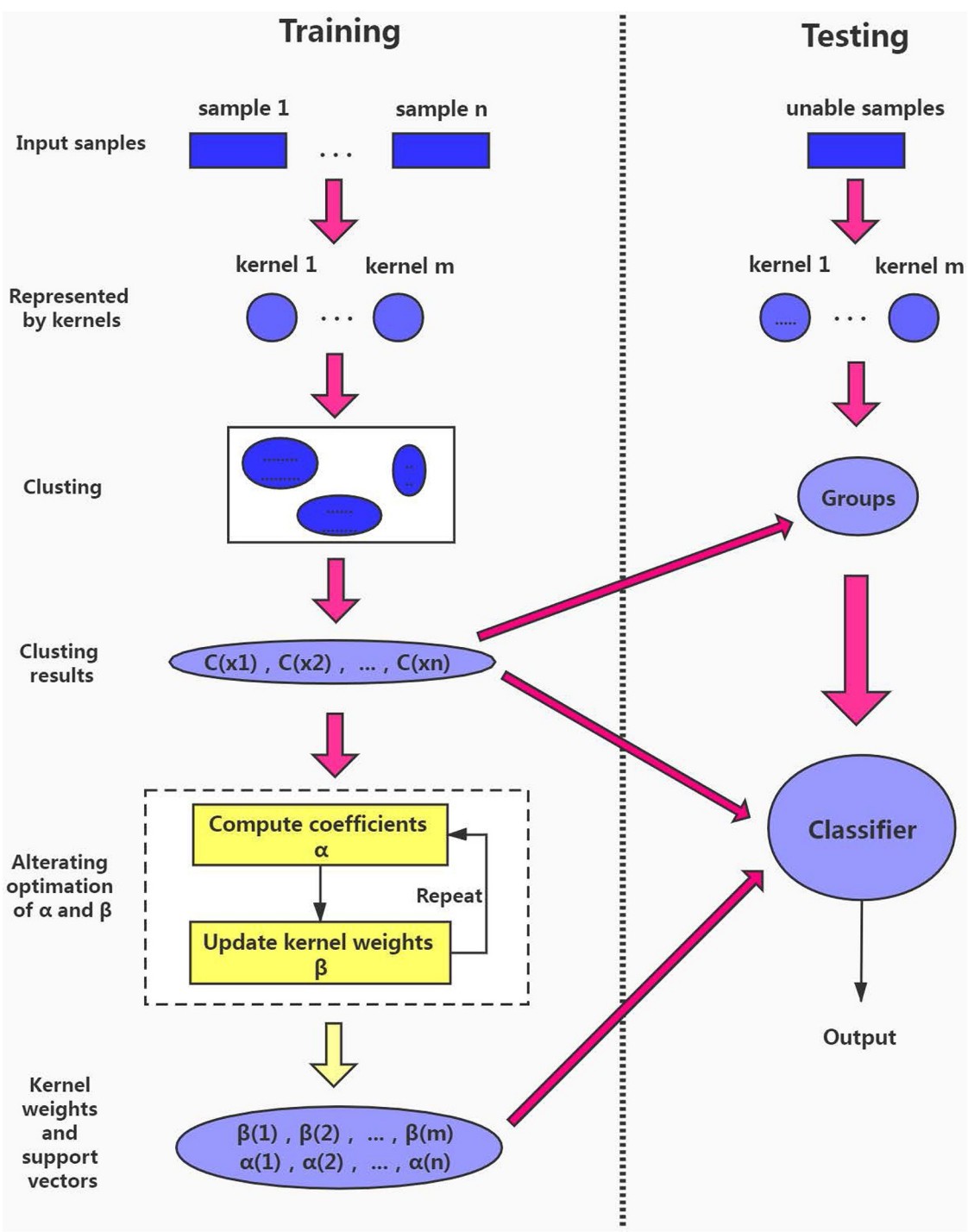

**Fig 5. The training and testing process in GLDMKL.**

Let $L$ have a partial bias of zero for the variable $\omega_m$, $b$, $\xi_i$, and we get the results Eqs (12)–(14).

$$\frac{\partial L}{\partial \omega_m} = 0 \Rightarrow \omega_m = \sum_{i=1}^{N} \beta_{c(\boldsymbol{x}_i),m}^{(l)} \phi_m(\boldsymbol{x}_i)\alpha_i y_i \qquad (12)$$

$$\frac{\partial L}{\partial b} = 0 \Rightarrow \sum_{i=1}^{N} \alpha_i y_i = 0 \tag{13}$$

$$\frac{\partial L}{\partial \xi_i} = 0 \Rightarrow C = \alpha_i + \gamma_i \tag{14}$$

Using the Eqs (12)–(14) to eliminate $\omega_m$, $b$ and $\xi_i$, we get the dual expression of the original optimization problem Eq (10) as **Theorem 1**.

$$\min_{\alpha} \quad \frac{1}{2} \sum_{i=1}^{N} \sum_{j=1}^{N} \alpha_i \alpha_j y_i y_j$$

$$\left( \sum_{m=1}^{M} \beta_{c(\mathbf{x}_i),m}^{(l)} \beta_{c(\mathbf{x}_j),m}^{(l)} k_m^{(l)}(\mathbf{x}_i, \mathbf{x}_j) \right) - \sum_{i=1}^{N} \alpha_i \tag{15}$$

$$\text{s.t.} \quad \sum_{i=1}^{N} \alpha_i y_i = 0, \quad 0 \leq \alpha_i \leq C \quad \forall i$$

Thus, we can rewrite the discriminant function Eq (9) to Eq (16).

$$f^{(l)}(\mathbf{x}) = sign \left( \sum_{i=1}^{N} \alpha_i y_i K^{(l)}(\mathbf{x}_i, \mathbf{x}) + b \right) \tag{16}$$

where

$$K^{(l)}(\mathbf{x}_i, \mathbf{x}) = \sum_{m=1}^{M} \beta_{c(\mathbf{x}_i),m}^{(l)} \beta_{c(\mathbf{x}),m}^{(l)} k_m^{(l)}(\mathbf{x}_i, \mathbf{x}) \tag{17}$$

The purpose of the *sign* function is to classify $f^{(l)}(\mathbf{x})$ as {-1, +1}.

**Definition 4** The objective function of the minimized Eq (15) is based on the fixed kernel weight $\beta$. If the kernel weight $\beta$ is added to the dual optimization problem, which is finally a max-min problem as Eq (18).

$$\max_{\beta} \min_{\alpha} J = \frac{1}{2} \sum_{i=1}^{N} \sum_{j=1}^{N} \alpha_i \alpha_j y_i y_j$$

$$\left( \sum_{m=1}^{M} \beta_{c(\mathbf{x}_i),m}^{(l)} \beta_{c(\mathbf{x}_j),m}^{(l)} k_m^{(l)}(\mathbf{x}_i, \mathbf{x}_j) \right) - \sum_{i=1}^{N} \alpha_i \tag{18}$$

$$\text{s.t.} \quad \sum_{i=1}^{N} \alpha_i y_i = 0, \quad 0 \leq \alpha_i \leq C \quad \forall i$$

$$\sum_{m} \left( \beta_{c(\mathbf{x}_i),m}^{(l)} \right)^{P} = 1 \; \beta_{c(\mathbf{x}_i),m}^{(l)} \geq 0 \; \forall i, m$$

$J$ is a multi-objective function with a coefficient $\alpha$ and a local kernel weight $\beta$. When $\beta$ is fixed, minimizing $J$ means minimizing global classification errors and maximizing the interval. When $\alpha$ is fixed, maximizing $J$ means maximizing sample similarity within the group while minimizing sample similarity between groups.

Similar to the canonical MKL, we alternately optimize $\alpha$ and $\beta$ to solve the max-min problem. In the first phase, we fix $\beta$ and optimize $\alpha$. It is easy to know that this problem is a

canonical SVM with a specific combined kernel that can be solved with the **Theorem 1**. In the second phase, we fixed $\alpha$ and optimized $\beta$, so $J$ can be rewritten as the Eq (19).

$$J(\beta) = \max \sum_{g=1}^{G} \sum_{g'=1}^{G} \sum_{m=1}^{M} \beta_{g,m}^{(l)} \beta_{g',m}^{(l)} S_m^{gg'}(\alpha^*) - \sum_{i=1}^{N} \alpha_i^* \tag{19}$$

where

$$S_m^{gg'}(\alpha^*) = \frac{1}{2} \sum_{\{i|c(\boldsymbol{x}_i)=g\}} \sum_{\{j|c(\boldsymbol{x}_j)=g'\}} \alpha_i^* \alpha_j^* y_i y_j k_m^{(l)}(\boldsymbol{x}_i, \boldsymbol{x}_j) \tag{20}$$

Where $\alpha^*$ is the optimization result of $\alpha$, and $\beta_{g,m}^{(l)}$ is the local kernel weight of the $m$-th kernel of group $g$ where sample **x** is located in layer $l$.

Note that the solution of $J(\beta)$ in the Eq (19) is independent of the latter term, which is equivalent to the problem of solving the Eq (21).

$$J(\beta) = \max_{\beta} \sum_{g=1}^{G} \sum_{g'=1}^{G} \sum_{m=1}^{M} \beta_{g,m}^{(l)} \beta_{g',m}^{(l)} S_m^{gg'}(\alpha^*) \tag{21}$$

Where $\beta_{g,m}^{(l)}$ represents the weight of the $m$-th basic kernel in group $g$ in the $l$ layer, $S_m^{gg'}(\alpha^*)$ is a shorthand for the dual formula that optimizes $\alpha$.

This is a quadratic non-convex problem. We know that solving the secondary planning problem requires expensive calculations. Inspired by [16, 39], we use the gated model to represent $\beta$.

**Definition 5** The gate function is designed as shown in Eq (22).

$$\beta_{g,m}^{(l)} = \frac{\exp\left(a_{g,m}^{(l)} v_{g,m}^{(l)} + b_{g,m}^{(l)}\right)}{\left(\sum_{m'=1}^{M} \exp p(a_{g,m'}^{(l)} v_{g,m'}^{(l)} + b_{g,m'}^{(l)})\right)^{\frac{1}{p}}} \tag{22}$$

Where $v_{g,m}^{(l)}$ is the kernel alignment [40] of the $m$-th kernel in sample group $g$ in layer $l$, $a$ and $b$ are parameters of the gate function.

So it can be calculated by the Eq (23).

$$v_{g,m}^{(l)} = \frac{\langle k_{g,m}^{(l)}, y_g y_g^T \rangle_F}{\sqrt{\langle k_{g,m}^{(l)}, k_{g,m}^{(l)} \rangle_F \langle y_g y_g^T, y_g y_g^T \rangle_F}} \tag{23}$$

where

$$\langle k_p^{(l)}, k_q^{(l)} \rangle_F = \sum_{i,j} k_p^{(l)}(\boldsymbol{x}_i, \boldsymbol{x}_j) k_q^{(l)}(\boldsymbol{x}_i, \boldsymbol{x}_j) \tag{24}$$

We transform the non-convex problem into a convex function problem through a gate function. In this way, the local minimum must be found by the gradient ascent method, so our method must converge.

We can observe $\sum_{m=1}^{M} (\beta_{g,m}^{(l)})^p = 1$ from the Eq (22), so $p$ norm is used to constrain $\beta_{g,m}^{(l)}$. We can optimize $p$ to change the sparseness of the kernels according to the datasets, thus changing the number of kernels used so that useful kernels can be fully utilized.

After the gate function is used to represent the local kernel weight, $J(\beta)$ becomes a convex function for $a$ and $b$. Therefore, we can optimize $a$ and $b$ by gradient ascent to maximize $J(\beta)$.

The bias of $J(\beta)$ for $a$ and $b$ are Eqs (25) and (26).

$$\frac{\partial J(\beta)}{\partial a_{g,m}^{(l)}} = 2\sum_{m'}^{M}\left(\sum_{i=1}^{G}\left(\beta_{i,m'}^{(l)}S_{m'}^{ig}(\alpha)\right)\beta_{g,m}^{(l)}v_{g,m}^{(l)}\left(\delta_m^{m'} - (\beta_{g,m'}^{(l)})^p\right)\right) \tag{25}$$

$$\frac{\partial J(\beta)}{\partial b_{g,m}^{(l)}} = 2\sum_{m'}^{M}\left(\sum_{i=1}^{G}\left(\beta_{i,m'}^{(l)}S_{m'}^{ig}(\alpha)\right)\beta_{g,m}^{(l)}\left(\delta_m^{m'} - (\beta_{g,m'}^{(l)})^p\right)\right) \tag{26}$$

If $m' = m$, then $\delta_m^{m'} = 1$, otherwise $\delta_m^{m'} = 0$. We update $a$ and $b$ with the gradient ascent method, then update $\beta_{g,m}^{(l)}$ with $a$ and $b$ as shown by the Eqs (27) and (28).

$$a_{g,m}^{(l)} + \lambda^t \frac{\partial J(\beta)}{\partial a_{g,m}^{(l)}} \rightarrow a_{g,m}^{(l)} \tag{27}$$

$$b_{g,m}^{(l)} + \mu^t \frac{\partial J(\beta)}{\partial b_{g,m}^{(l)}} \rightarrow b_{g,m}^{(l)} \tag{28}$$

Where $\lambda^t$ and $\mu^t$ are the step sizes, which can be obtained by a line search method as [41] or fixed as a small constant.

In this way, optimizing $\alpha$ and $\beta$ alternately until certain termination criteria are met. We use the duality gap as the termination criterion, which is written as shown in Eq (29).

$$\max_m \sum_{i=1}^{N}\sum_{j=1}^{N}\alpha_i\alpha_j y_i y_j k_m^{(l)}(\boldsymbol{x}_i,\boldsymbol{x}_j) -$$
$$\sum_{i=1}^{N}\sum_{j=1}^{N}\alpha_i\alpha_j y_i y_j\left(\sum_{m=1}^{M}\beta_{c(\boldsymbol{x}_i),m}^{(l)}\beta_{c(\boldsymbol{x}_j),m}^{(l)}k_m^{(l)}(\boldsymbol{x}_i,\boldsymbol{x}_j)\right) \leq \varepsilon \tag{29}$$

Where $\varepsilon$ is the preset tolerance threshold.

## GLDMKL learning algorithm

Given a set of training data $D = \{(\mathbf{x}_i, y_i)|i = 1, 2, \ldots, n\}$, where $\boldsymbol{x}_i \subseteq \mathbb{R}^d$ is the sample feature vector, $y_i \in \{-1, +1\}$ is the sample class label. Our goal is to train deep multiple kernel networks and the SVM classifiers from labeled training data.

After the training samples are grouped by multiple kernel k-means clustering, the original local combined kernel form of the first layer is the Eq (30).

$$K^{(1)}(\boldsymbol{x}_i,\boldsymbol{x}_j) = \sum_{m=1}^{M}\beta_{c(\boldsymbol{x}_i),m}^{(1)}\beta_{c(\boldsymbol{x}_j),m}^{(1)}k_m^{(1)}(\boldsymbol{x}_i,\boldsymbol{x}_j) \tag{30}$$

Where $\beta_{c(\boldsymbol{x}),m}^{(1)}$ represents the weight of the $m$-th kernel of the group $c(\mathbf{x})$ where sample $\mathbf{x}$ is located in the first layer, $k_m^{(1)}(\boldsymbol{x}_i,\boldsymbol{x}_j)$ denotes the $m$-th kernel between sample $\mathbf{x}_i$ and $\mathbf{x}_j$ in the first layer.

To make it easier to express the relationship between the combined kernels in deep multiple kernel learning, we simplify the local combined kernel of the group $g$ in the first layer into the

form Eq (31).

$$K_g^{(1)} = \sum_{m=1}^{M} \beta_{g,m}^{(1)} k_{g,m}^{(1)} \tag{31}$$

The next step is the combined kernel relationship between the previous layer and the next layer. Before this, we must understand the principle of deep kernel learning.

**Definition 6** The following is the principle formula of deep kernel learning.

$$K^{(L)}(\boldsymbol{x}_i, \boldsymbol{x}_j) = \Phi^{(L)}\left( \ldots \Phi^{(1)}(\boldsymbol{x}_i) \right) \cdot \Phi^{(L)}\left( \ldots \Phi^{(1)}(\boldsymbol{x}_j) \right) \tag{32}$$

Where $\mathbf{x}_i$ and $\mathbf{x}_j$ are input feature vectors, $\Phi^{(L)}$ is a feature mapping function applied $L$ times, $K^{(L)}$ represents the final layer kernel which is the combined kernel of the SVM classifier when reaching the cutoff condition.

From **Definition 6**, the classifier models of the 1 to $L$ layer structures are used for classification, the errors are calculated and the accuracies are obtained. As long as the accuracy is continuously unchanged for several layers, the number of layers can be stopped to increase.

**Definition 7** The following is the derivation Eq (33) for deep multiple kernel learning.

$$\begin{aligned} K^{(l)} = \quad & \left\{ \sum_{m=1}^{M} \beta_m^{(l)} k_m^{(l)}(K^{(l-1)})| \right. \\ & \left. \beta_m^{(l)} \geq 0, \ m = 1, \ldots, M, \ l = 2, \ldots, L \right\}, \\ & K^{(1)} = \sum_{m=1}^{M} \beta_m^{(1)} k_m^{(1)} \end{aligned} \tag{33}$$

Where $K^{(l)}$ is the output value of the combined kernel in layer $l$.

Our architecture is based on the theory of local learning and performs GLDMKL. The output value of the combined kernel in layer $l-1$ is used as an input to the combined kernel in layer $l$, and the local deep multiple kernel learning derivation equation in group $g$ is as Eq (34).

$$K_g^{(l)} = \sum_{m=1}^{M} \beta_{g,m}^{(l)} k_{g,m}^{(l)}(K_g^{(l-1)}) \tag{34}$$

Where $K_g^{(l)}$ is the output value of the combined kernel in group $g$ in layer $l$, $\beta_{g,m}^{(l)}$ indicates the weight of the $m$-th basic kernel in group $g$ in layer $l$, $k_{g,m}^{(l)}$ represents the $m$-th basic kernel in group $g$ in layer $l$. The final decision function of the proposed framework in group $g$ in layer $l$ is defined as Eq (35).

$$f_g^{(l)}(\boldsymbol{x}) = \sum_{i=1} \alpha_i y_i K_g^{(l)} + b \tag{35}$$

Adaptive deep multiple kernel learning is performed in each group, there needs to perform a decision function for predicting classification accuracy in each layer. And note that the final layer of the decision function may be different due to the independent prediction in each group.

In layer $l-1$, the idea of **Definition 1—Definition 5** in GLDMKL is utilized, and taking turns to optimize the support vector parameter $\alpha$ and the local kernel weight $\beta_{g,m}^{(l-1)}$. The gradient ascent method is used to update the weights, and then the fixed weights are used to

calculate the support vector parameters until the cutoff condition is satisfied, and the classification accuracy of the local SVM classifier in each group is obtained.

Then we will use the output value of the combined kernel in layer $l - 1$ as an input to the combined kernel in layer $l$, and continue to repeat the idea of the GLDMKL learning algorithm. The classification accuracy is needed to calculate in each layer at the same time. The highest classification accuracy remains unchanged in successive layers; then the local deep multiple kernel learning is stopped immediately.

To obtain a coefficient that minimizes the real risk of the decision function, the gradient ascent is used to optimize the local kernel weight $\beta_{g,m}^{(l)}$. In the local space in layer $l$, $J(\beta)$ is used from Eq (21).

We can maximize $J(\beta)$ by using an iterative process of gradient ascent. First, we calculate the gradient of each weight in each local space in each layer as the Eq (36).

$$\nabla J(\beta) = \left\{ \begin{array}{l} \dfrac{\partial J(\beta)}{\partial \beta_{1,1}^{(1)}}, \ldots, \dfrac{\partial J(\beta)}{\partial \beta_{1,m}^{(1)}}, \ldots, \dfrac{\partial J(\beta)}{\partial \beta_{g,1}^{(1)}}, \ldots, \dfrac{\partial J(\beta)}{\partial \beta_{g,m}^{(1)}}, \\[3mm] \dfrac{\partial J(\beta)}{\partial \beta_{1,1}^{(2)}}, \ldots, \dfrac{\partial J(\beta)}{\partial \beta_{1,m}^{(2)}}, \ldots, \dfrac{\partial J(\beta)}{\partial \beta_{g,1}^{(2)}}, \ldots, \dfrac{\partial J(\beta)}{\partial \beta_{g,m}^{(2)}}, \\[2mm] \cdots\cdots\cdots \\[2mm] \dfrac{\partial J(\beta)}{\partial \beta_{1,1}^{(L)}}, \ldots, \dfrac{\partial J(\beta)}{\partial \beta_{1,m}^{(L)}}, \ldots, \dfrac{\partial J(\beta)}{\partial \beta_{g,1}^{(L)}}, \ldots, \dfrac{\partial J(\beta)}{\partial \beta_{g,m}^{(L)}} \end{array} \right\} \tag{36}$$

Then, gradient ascent is used to update all local kernel weights of local deep multiple kernel learning in each layer as the Eq (37).

$$\begin{array}{c} \beta_{g,m}^{(l)} + \eta \nabla J(\beta) = \beta_{g,m}^{(l)} \ \ \beta_{g,m}^{(l)} \geq 0 \\[2mm] m = 1, \ldots, M \ \ l = 1, \ldots, L \end{array} \tag{37}$$

Where $\eta$ is the step size, so we just need $\nabla J(\beta)$ to figure out the local kernel weights, and $\nabla J(\beta)$ is obtained from the gate function in **Definition 5**. The specific operation process can be found in the Eqs (25)–(28).

In the model learning algorithm, we can apply the alternating optimization algorithm used in GLDMKL to learn the decision function coefficient $\alpha$ and all local kernel weights $\beta_{g,m}^{(l)}$. This can be done: 1) fixing $\beta_{g,m}^{(l)}$ and solving $\alpha$ using the normal method; 2) fixing $\alpha$ and using gradient ascent to solve $\beta_{g,m}^{(l)}$ until the optimization deadline is met.

The classification accuracy is evaluated in each layer to determine whether the growth of the model layer is stopped. If the highest classification accuracy does not change in the fixed number of layers, stopping layers growing.

The entire process of our model learning algorithm is described in the *Algorithm*1.

According to the *Algorithm*1, a layer of the GLDMKL architecture is constructed from an iteration of step 5 to 15. In the *Algorithm*1, $i$ represents the current number of layers, and $j$ represents the number of layers that maximum accuracy $A_m$ remains unchanged, and $j$ is a Judge condition for stopping the growth of the layer. Step 7 to 10 indicate that local multiple kernel learning with the lp norm is performed in the current layer $l_i$, and $\alpha$ and $\beta$ are optimized in turn until the cutoff condition is reached. At the same time, the classification accuracy $A_{cc}$ of the SVM classifier in each group is calculated, and the highest classification accuracy $A_m$ is updated. If the best accuracy does not change in the fixed number of layers, then the number

of layers should be stopped. And if the number of iterations exceeds the preset maximum number of iterations, the iteration is stopped.

**Algorithm 1** Group-based local adaptive deep multiple kernel learning algorithm

```
Input: D: Dataset
  m: Number of candidate kernels
  k_m: Initial parameters of each kernel
  g: Number of groups
  l_acc: Maxmum number of layers in which the best accuracy does not
change
  L: Maxmum number of iteration layers
Output: Final classification model M
1: Randomly select 50 percent of samples from the entire dataset D as
training samples D_T;
2: Divide the training samples D_T into g groups with Clustering;
3: Initialize best accuracy A_m = 0;
4: Initialize current iteration i = 0 and flag j = 0;
5: repeat
6:   Use grid search method to adjust the initial parameters k_m;
7:   repeat
8:     Initialize gate model parameters a, b with small random numbers;
9:     Rotation optimize α and β with GLMKL algorithm in layer l_i;
10:  until meet the termination criterion of Eq (28);
11:  Update the best accuracy A_m;
12:  If A_m does not change then
     j++;
13:  The output of the combined kernel in the previous layer l_i is
used as the input of the combined kernel in the next layer l_{i+1};
14:  i++;
15:  untill (i >= L or j >= l_acc).
```

## Experiments

### Setup

The main implementation of GLDMKL is written in Python, and related algorithms can call library functions. All experiments were run on a PC with a 2.2 GHz Intel Core i5-5200U CPU and 12 GB RAM and win7 operating system and GTX 1080 GPU server.

In kernel settings, RBF and polynomial kernels are usually the most commonly used functions for MKL methods. In our work, we have selected four basic kernels and Arc-cosine kernel.The detailed kernel parameter settings are shown in Table 1.

In the comparison experiment, the other five classical comparison methods are shown in Table 2.

To simplify the experiment, we initialize the maximal iteration number *maxIter* in parameter optimization to 100, the maximum layer number *L* of the model is 20, and the maximum number of layers $l_{acc}$ with the best precision unchanged is set to 3 layers.

### Data

**UCI datasets.**   We perform a set of extended experiments to evaluate the performance of the proposed GLDMKL algorithm in the classification task with small sample sizes and low dimensions. Several algorithms have been tested on six real-world datasets: Liver, Breast, Sonar, Australian, German, Monk. Table 3 gives a detailed description of the usage datasets.

Among them, Liver and German are datasets with relatively complex data. Sonar and Australian are datasets with relatively simple data. Breast and Monk are datasets with very simple data.

**Table 1. Kernel parameters setting.**

| Kernel | Equation | Parameters |
|---|---|---|
| Polynomial | $k(\mathbf{x}_i, \mathbf{x}_j) = (\gamma^* <\mathbf{x}_i, \mathbf{x}_j> + c)^n$ | $\gamma = 1.2\ c = 2.1\ n = 1$ |
| Laplacian | $k(\pmb{x}_i, \pmb{x}_j) = exp\left(-\frac{\|x_i - x_j\|}{\sigma}\right)$ | $\sigma = 1.2$ |
| Tanh | $k(\mathbf{x}_i, \mathbf{x}_j) = tanh(\beta^* <\mathbf{x}_i, \mathbf{x}_j> + \theta)$ | $\beta = 1.2\ \theta = 2.1$ |
| RBF | $k(\pmb{x}_i, \pmb{x}_j) = exp\left(-\frac{\|x_i - x_j\|^2}{2\sigma^2}\right)$ | $\sigma = 0.9$ |
| Arc-cosine [42] | $k_n(\pmb{x}_i, \pmb{x}_j) = \frac{1}{\pi} \| \pmb{x}_i\|^n * \| \pmb{x}_j \|^n J_n(\theta)$ $J_n(\theta) = (-1)^n (\sin\theta)^{2n+1} \left(\frac{1}{\sin\theta}\frac{\partial}{\partial\theta}\right)^n \left(\frac{\pi - \theta}{\sin\theta}\right)$ $\theta = \cos^{-1}\left(\frac{< \pmb{x}_i, \pmb{x}_j >}{\| \pmb{x}_i \| * \| \pmb{x}_j \|}\right)$ | $\| a \|\quad and\quad \| b \|$ $are\ L_0\ norm$ $n = 0$ |

**Table 2. Five classical comparison methods.**

| Methods | Detail | Year |
|---|---|---|
| 2LMKL | Zhuang proposed a two-layer multiple kernel learning algorithm [12] | In 2011 |
| DMKL | Deep multiple kernel learning algorithm proposed by strobl [13] | In 2013 |
| MLMKL | Multi-layer multiple kernel learning algorithm for backpropagation proposed by Rebai [14] | In 2016 |
| SA-DMKL | Adaptive deep multiple kernel learning algorithm proposed [15] | In 2019 |
| DWS-MKL | Depth-width-scaling multiple kernel learning algorithm proposed [20] | In 2020 |

**Caltech-256 datasets.** Caltech-256 is an image object recognition dataset containing 30,608 images and 256 object categories, each has at least 80 images. We select the Caltech-256 dataset to evaluate the performance of our GLDMKL approach in classification tasks with large sample sizes and high dimensions.

In our experiments, we randomly select five types of data with similar shapes: bowling-ball, car-tire, desk-globe, roulette-wheel, and sunflower-101.

Before the training of the image datasets, feature extraction is required, and FFT is used as the descriptor of the image dataset. Removing irrelevant image features, and then simple processing of image features is carried out to facilitate the formation of features.

**Data pretreatment.** First, it needs to preprocess for each dataset. Samples are normalized such that the characteristic numbers are in the range of 0 to 1, thereby preventing overflow of data manipulation during the experiment. Next, we randomize samples and divide them into two halves: (1) 50% of the examples are used for training (establishing a deep multiple kernel

**Table 3. Selected datasets in UCI.**

| Datasets | Dimensions | Samples |
|---|---|---|
| Liver | 6 | 345 |
| Breast | 9 | 286 |
| Sonar | 60 | 208 |
| Australian | 14 | 690 |
| German | 20 | 1000 |
| Monk | 6 | 432 |

model with the best parameters), and (2) the remaining 50% is used as test data (evaluating the performance of the resulting model). Finally, the six UCI datasets and five image datasets are used to train our model and test our classification model with the same test datasets. To ensure the reliability of the data, we run ten times for each dataset and take the best classification results.

## Metrics

Common classification performance metrics are the accuracy, the generalization ability, the training time, etc. Generalization ability is not easily measured by data, so it is not used. And because the experimental environment is different in different methods, the superiority of our method cannot be well reflected, so the training time is not used. However, the classification accuracy can intuitively reflect the classification performance of a method, and it is the best performance metric, not affected by the experimental environment. Therefore, accuracy is used as a performance indicator to compare with other methods.

According to Eq (38), the learning performance of the GLDMKL method is evaluated according to the test accuracy. And Eq (38) represents the ratio of the number of correctly classified samples to the total number of samples.

$$Accuracy = \frac{TP + TN}{N} \tag{38}$$

Where $TP$ is the number of true positives, $TN$ is the number of true negatives and $N$ is the total number of samples in the test datasets.

## The experimental results in UCI datasets

We have successfully completed three experiments. The first is to compare the classical DMKL method with our GLDMKL method in classification performance. The second experiment shows the effect of the number of clustering groups in our GLDMKL method on classification performance. The last experiment shows the effect of layers on the classification performance.

**Comparison of classification performance.** The purpose of this experiment is to evaluate the performance of the DMKL algorithm using our GLDMKL method and other classical DMKL methods on the UCI dataset. Therefore, we evaluate the following algorithms: 2LMKL, DMKL, MLMKL, SA-DMKL, DWS-MKL and our GLDMKL. Table 4 shows the detailed results of the classification performance for the different algorithms. Among them, data with the highest classification accuracy for the same dataset is highlighted in bold.

By comparing the results among SA-DMKL, DWS-MKL and other DMKL methods (2LMKL, DMKL and MLMKL), we find that SA-DMKL and has higher performance than other DMKL methods. For example, SA-DMKL is superior to other algorithms on three

**Table 4. Comparison of best classification performance(%).**

| Datasets | Algorithms | | | | | |
|---|---|---|---|---|---|---|
| | 2LMKL | DMKL | MLMKL | SA-DMKL | DWS-MKL | GLDMKL |
| Liver | 63.43 | 69.01 | 71.80 | 75.65 | 74.85 | **80.00** |
| Breast | 96.53 | 96.59 | 97.21 | 91.92 | 97.08 | **99.99** |
| Sonar | 83.75 | 83.94 | 83.84 | 89.42 | 84.79 | **99.99** |
| Australian | 82.11 | 84.40 | 85.42 | 82.03 | 85.51 | **99.99** |
| German | 72.22 | 72.02 | 75.06 | 78.50 | 73.60 | **80.52** |
| Monk | 96.26 | 96.62 | 96.89 | 97.55 | 99.07 | **99.99** |

datasets: Liver, Sonar, and German. DWS-MKL is superior to other algorithms on three datasets: Breast, Australian, and Monk. This shows that the adaptive layer structure can achieve higher performance than the fixed structure.

It can be seen from Table 4 that our GLDMKL method has better classification performance on the above six datasets than other methods. Moreover, classification accuracies in some datasets reach 99.99%, which shows that the idea of our local adaptive deep multiple kernel learning based on grouping is feasible and the effect is also significant. In other methods, 2LMKL shows the worst performance on the Liver, Sonar, and Monk datasets, while SA-DMKL shows the worst performance on the Breast dataset with very simple data and the Australian dataset with relatively simple data. Therefore, these results show that our GLDMKL method is superior to the classical DMKL method and can be widely adapted to a variety of datasets.

**The effect of group numbers on classification performance.**   In this experiment, we explore the effect of the number of groups on the classification performance in our method. To simplify the experiment, we specially extract clusters into 1 group, 2 groups, 5 groups, 7 groups, and 10 groups as experimental comparisons. As can be seen from Fig 6, the more the number of groups in most datasets, the higher the classification performance will be. Especially for the Monk dataset, the improvement of classification performance is most obvious for multi-grouping, indicating that the dataset with very simple data and is consistent with the actual situation.

For the German dataset with relatively complex data, the classification accuracy increases with the number of groups increasing. For the Sonar dataset, when the number of groups reaches 5 or 7, the classification accuracy is the highest. And increasing the number of groups, it will be greatly reduced, indicating that the dataset is not suitable for grouping too much. So it is the best choice to group into 5 or 7 in the sonar dataset with relatively simple data.

**The effect of layer numbers on classification performance.**   In this experiment, we evaluate the classification performance of different DMKL methods in each layer: DMKL, MLMKL, SA-DMKL, DWS-MKL and our GLDMKL to explore the effect of layers on classification

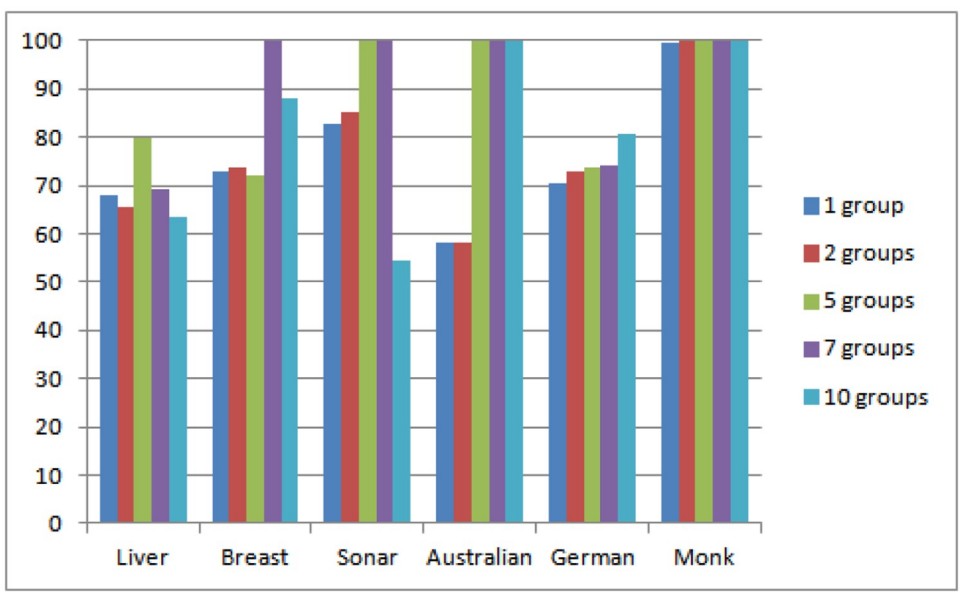

**Fig 6. Performance comparison(%) of different groups.**

**Table 5. Comparison of classification performance(%) at each layer.**

| Dataset | Algorithms | | | | | | | | | | | | |
|---|---|---|---|---|---|---|---|---|---|---|---|---|---|
| | DMKL | | | MLMKL | | | SA-DMKL | DWS-MKL | | | GLDMKL | | |
| | 1-layer | 2-layer | 3-layer | 1-layer | 2-layer | 3-layer | $*$-layer | 1-layer | 2-layer | 3-layer | 1-layer | 2-layer | 3-layer |
| Liver | 69.53 | 69.53 | 69.01 | 69.53 | 70.17 | 71.80 | 75.65 | 74.85 | 72.51 | 72.51 | 74.00 | **80.00** | 76.92 |
| Breast | 94.92 | 96.56 | 96.59 | 96.89 | 96.83 | 97.21 | 91.92 | 96.78 | 97.08 | 97.08 | **99.99** | 92.59 | 72.73 |
| Sonar | 82.98 | 84.13 | 83.94 | 83.94 | 84.23 | 83.84 | 89.42 | 84.76 | 84.76 | 84.76 | **99.99** | **99.99** | 74.04 |
| Australian | 83.56 | 84.26 | 84.40 | 85.13 | 85.56 | 85.42 | 82.03 | 84.35 | 84.64 | 85.51 | **99.99** | **99.99** | **99.99** |
| German | 70.56 | 71.48 | 72.02 | 73.80 | 74.56 | 75.06 | 78.50 | 72.00 | 72.00 | 73.60 | **80.52** | 74.25 | 79.01 |
| Monk | 96.25 | 96.52 | 96.62 | 96.89 | 96.62 | 96.89 | 97.55 | 99.07 | 99.07 | 99.07 | **99.99** | **99.99** | 99.54 |

performance. The DMKL, MLMKL, DWS-MKL and GLDMKL methods all train up to three layers to analyze and determine the effectiveness of using multi-layer structures. Since predecessors only have counted the experimental results of three layers, we also write the experimental results of three layers for comparison to facilitate the comparison experiment. And the SA-DMKL method does not perform a comparison experiment about the number of layers, so the number of layers is assumed to be $*$-layer. Table 5 shows the detailed results of the different method classifications. Among them, data with the highest classification accuracy for the same dataset is highlighted in bold.

First, by comparing the results between DMKL and MLMKL methods, we find that DMKL and MLMKL with a multi-layer structure can slightly improve the classification accuracy, but it is not obvious, and even sometimes the classification accuracy is reduced. For the DMKL method, as the number of layers increases, the classification accuracy will decrease in the Liver dataset with relatively complex data. However, we notice that increasing the number of layers has still a certain impact on the improvement of classification accuracy.

Secondly, as can be seen from Table 5, the accuracy increases with the number of layers in the DMKL and MLMKL methods. Moreover, Our GLDMKL method improves the classification accuracy more obviously than the other two methods. For the Australian dataset with relatively simple data, the classification accuracy is the highest at each layer. The classification accuracy is also improved obviously in the Liver and German datasets with relatively complex data.

In the end, by comparing our GLDMKL method with other classical DMKL methods, our method can complete the classification task in a shorter time than other methods. For the Breast and German dataset, there achieves the best classification accuracy in the first layer and the improvement is very obvious. It can be seen that our GLDMKL method plays the role of shortening the number of layers, which will greatly shorten the training time and makes it achieve the best classification effect faster. Perhaps because these datasets are too simple, our method can be faster and more accurate in the classification task.

## The experimental results in Caltech-256 datasets

We have successfully completed five experiments. The first is the classification performance experiment for each layer when the bowling-ball class is positive. The second shows the classification performance experiment for each layer when the car-tire class is positive. The third shows the classification performance experiment for each layer when the desk-globe class is positive. The fourth experiment shows the effect of layers on the classification performance when the roulette-wheel class is positive. The last experiment shows the effect of layers on the classification performance when the sunflower-101 class is positive. To simplify the

**Table 6. Classification performance(%) at each layer where the bowling-ball class is positive.**

| Layers | Groups | | | | |
|---|---|---|---|---|---|
| | 1 group | 2 groups | 5 groups | 7 groups | 10 groups |
| 1-layer | 87.22 | 86.55 | **90.00** | **90.00** | 80.00 |
| 2-layer | 87.51 | 86.55 | 86.97 | **90.00** | 61.76 |
| 3-layer | 87.01 | 86.63 | 86.97 | **90.00** | 85.29 |
| 4-layer | 89.05 | **90.00** | 69.27 | 86.67 | 63.73 |
| 5-layer | 89.79 | **90.00** | 84.90 | 87.14 | 75.49 |

**Table 7. Classification performance(%) at each layer where the car-tire class is positive.**

| Layers | Groups | | | | |
|---|---|---|---|---|---|
| | 1 group | 2 groups | 5 groups | 7 groups | 10 groups |
| 1-layer | 89.01 | 87.70 | 87.88 | 88.49 | **89.99** |
| 2-layer | 87.01 | 80.80 | 82.00 | 88.49 | 71.14 |
| 3-layer | 86.52 | 80.80 | 82.00 | 88.49 | 71.14 |
| 4-layer | 87.51 | 80.80 | 82.00 | 88.49 | **89.99** |
| 5-layer | 88.51 | 80.80 | 82.00 | 88.49 | **89.99** |

experiment, we also extract clusters into 1 group, 2 groups, 5 groups, 7 groups, and 10 groups as experimental comparisons. To describe the effect of layers, we write the classification accuracy results in the first five layers. Tables 6–10 show the detailed results. Among them, data with the highest classification accuracy is highlighted in bold. The invalid accuracy data after reaching the cut-off condition is marked in underlined.

**Table 8. Classification performance(%) at each layer where the desk-globe class is positive.**

| Layers | Groups | | | | |
|---|---|---|---|---|---|
| | 1 group | 2 groups | 5 groups | 7 groups | 10 groups |
| 1-layer | 89.36 | 70.00 | 88.63 | 85.63 | 84.12 |
| 2-layer | 55.81 | 71.63 | 89.32 | 85.63 | 84.12 |
| 3-layer | 83.97 | 71.63 | 89.32 | 85.00 | 84.05 |
| 4-layer | 37.74 | **90.00** | 89.32 | 85.00 | 84.05 |
| 5-layer | 55.92 | 85.43 | 87.40 | **90.00** | 86.43 |

**Table 9. Classification performance(%) at each layer where the roulette-wheel class is positive.**

| Layers | Groups | | | | |
|---|---|---|---|---|---|
| | 1 group | 2 groups | 5 groups | 7 groups | 10 groups |
| 1-layer | 88.52 | 79.50 | **89.88** | 80.00 | 88.86 |
| 2-layer | 89.37 | 88.02 | 88.13 | 72.36 | 88.86 |
| 3-layer | 89.37 | **89.88** | 88.13 | 72.36 | 80.68 |
| 4-layer | 89.37 | 83.30 | 88.50 | 72.36 | 80.68 |
| 5-layer | 89.37 | **89.88** | 88.50 | 72.36 | 80.68 |

**Table 10. Classification performance(%) at each layer where the sunflower-101 class is positive.**

| Layers | Groups | | | | |
|---|---|---|---|---|---|
| | **1 group** | **2 groups** | **5 groups** | **7 groups** | **10 groups** |
| 1-layer | 87.57 | 87.11 | **90.00** | 88.57 | **90.00** |
| 2-layer | 85.56 | 81.48 | **90.00** | 88.57 | **90.00** |
| 3-layer | 85.56 | 85.59 | **90.00** | 84.29 | **90.00** |
| 4-layer | 85.56 | 85.59 | **90.00** | 84.29 | **90.00** |
| 5-layer | 85.56 | 85.59 | **90.00** | 84.29 | **90.00** |

In the first experiment, a bowling-ball class is used as a positive class. The highest classification accuracy of 90.00% can be achieved in 5 groups and 7 groups, and the highest classification accuracy can be maintained in the first three layers. It can be seen that our method can be quickly distinguished from other classes in 5 and 7 groups. In 2 groups, with the increase of layers, the classification accuracy reaches the maximum in the fourth layer and then remains unchanged. It can be concluded that more layers will be needed to achieve the highest classification accuracy when the number of groups is small. And when the number of groups is large, the classification accuracy may decline, indicating that 5 and 7 groups are suitable for classification when the bowling-ball class is positive.

In the second experiment, a car-tire class is used as a positive class. The highest classification accuracy of 89.99% can be achieved in 10 groups. In 7 groups, the classification accuracy will not change with the number of layers and will remain as 88.49%. The experiment shows that only three layers are needed to reach the cutoff condition, indicating that the car-tire class can be quickly distinguished from other classes. As the number of groups increases, the highest classification accuracy in each group tends to increase. As we can be seen that it is easier to achieve the highest classification accuracy when the number of groups is relatively large.

In the third experiment, a desk-globe class is used as a positive class. The highest classification accuracy of 90.00% can be achieved in 2 groups and 7 groups. In 2 groups, the highest classification accuracy is achieved in the fourth layer and then decreases. In 7 groups, the highest classification accuracy is achieved in the fifth layer. In 5 groups, the classification accuracy in each layer remains relatively high. In 10 groups, with the increase in the number of layers, the classification accuracy tends to increase and cannot reach the highest in the fifth layer. It can be concluded that more layers will be needed to achieve the highest classification accuracy when the number of layers increases.

In the fourth experiment, a roulette-wheel class is used as a positive class. The highest classification accuracy of 89.88% can be achieved in 2 groups and 5 groups. In 5 groups, the highest classification accuracy in the first three layers is always 89.88% and the growth of layers can be stopped. In 2 groups, the highest classification accuracy is achieved in the third layer and remains unchanged in three successive layers, having met the cutoff conditions. When the number of layers is large, only three layers can reach the cutoff condition, indicating that the increase of layers will accelerate the process of classification, but it will not necessarily achieve the highest classification accuracy.

In the last experiment, a sunflower-101 class is used as a positive class. The highest classification accuracy of 90.00% can be achieved in 5 groups and 10 groups. And the classification accuracy in each layer is 90.00% in 5 groups and 10 groups, so there only needs three layers can reach the cutoff condition. It can be seen that 5 and 10 groups are suitable for classification when the sunflower-101 class is positive.

## Discussion

We have done three experiments on UCI datasets. The first experiment can show that the classification performance of our GLDMKL method is better than other classical DMKL methods. The second experiment shows that the number of groups has a certain effect on most datasets and different datasets had different effects. The third experiment shows that our GLDMKL method can shorten the number of model layers and reduce training and prediction time.

We also have done five experiments on Caltech-256 datasets. When the bowling-ball class, desk-globe class or sunflower-101 class is used as a positive class, the highest classification accuracy is 90.00%. When the car-tire class is used as a positive class, the highest classification accuracy is 89.99%. When the roulette-wheel class is used as a positive class, the highest classification accuracy is 89.88%. In conclusion, different classification tasks will have different highest classification accuracy.

## Validation

Our GLDMKL learning method is superior to other classical DMKL methods in classification accuracy, but there are also certain potential problems.

Firstly, clustering into multiple groups is required for training the model and for testing samples. And the testing samples maybe not tested in the corresponding grouping model, which may lead to lower classification accuracy. So the testing process must be repeatedly performed to get the best results.

Also, there is a probability in which group samples fall into, and the sample does not necessarily belong to this group. So the probabilistic grouping can be added in our model which is more suitable for the actual situation. The probabilistic grouping is the next major point we need to overcome.

Furthermore, if the lp norm is not handled properly, it will increase the sparseness of the kernel and will also result in lower classification accuracy.

Moreover, the stability of our model also needs to be considered, such as analyzing from the mean and standard deviation of the classification accuracy.

In the end, if samples are very simple, we can directly treat them as a group. And there is the same effect as classical DMKL methods.

## Conclusion

This paper proposes a new group-based local adaptive deep multiple kernel learning method (GLDMKL) with lp norm. Our GLDMKL architecture consists of two parts: multiple kernel k-means clustering and local adaptive deep multiple kernel learning. Furthermore, the layer is not fixed and will grow adaptively based on the actual datasets. Our model learning algorithm utilizes deep kernel learning to build a local deep multiple kernel learning model layer by layer. In our model learning algorithm, we can divide samples into groups according to the multiple kernel k-means clustering algorithm. And the SVM model parameters and local kernel weights corresponding groups were optimized in turn to fit the model. The hyperparameters of basic kernels are adjusted by the grid search method. According to the local kernel weight, the proportion of basic kernels in the combined kernel at each layer is changed. And the weight constraint with the lp norm is proposed. So the local kernel weights are adjusted with the lp norm and further controlling the sparseness of the kernel. The experimental results show that our GLDMKL method can test samples in the corresponding grouping model, and achieve better performance than other classical DMKL methods on a wide range of datasets. In future work, we plan to integrate more learning technologies into our GLDMKL methods, such as localization optimization, changes in distance definitions, deep kernel model

optimization, changes in data dimensions. And our GLDMKL approach will be implemented in embedded systems.

## Author Contributions

**Conceptualization:** Shengbing Ren, Fa Liu.

**Data curation:** Fa Liu, Xian Feng.

**Funding acquisition:** Shengbing Ren.

**Investigation:** Shengbing Ren, Fa Liu, Weijia Zhou.

**Methodology:** Shengbing Ren, Fa Liu, Xian Feng.

**Project administration:** Shengbing Ren.

**Software:** Fa Liu.

**Supervision:** Shengbing Ren.

**Validation:** Shengbing Ren, Weijia Zhou, Xian Feng, Chaudry Naeem Siddique.

**Writing – original draft:** Fa Liu.

**Writing – review & editing:** Shengbing Ren, Weijia Zhou, Xian Feng, Chaudry Naeem Siddique.

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
