## [Decision Letter · Decision Letter 0]

6 Jul 2020

PONE-D-20-09329

Group-based Local Adaptive Deep Multiple Kernel Learning with Lp Norm

PLOS ONE

Dear Dr. Ren,

Thank you for submitting your manuscript to PLOS ONE. After careful consideration, we feel that it has merit but does not fully meet PLOS ONE’s publication criteria as it currently stands. Therefore, we invite you to submit a revised version of the manuscript that addresses the points raised during the review process.

We look forward to receiving your revised manuscript.

Kind regards,

Robertas Damasevicius

Academic Editor

PLOS ONE

Journal Requirements:

3. We note that Figures 6, 7, 8, 9, 10 in your submission contain copyrighted images. All PLOS content is published under the Creative Commons Attribution License (CC BY 4.0), which means that the manuscript, images, and Supporting Information files will be freely available online, and any third party is permitted to access, download, copy, distribute, and use these materials in any way, even commercially, with proper attribution. For more information, see our copyright guidelines: http://journals.plos.org/plosone/s/licenses-and-copyright.

3.1.         You may seek permission from the original copyright holder of Figures 6, 7, 8, 9, 10 to publish the content specifically under the CC BY 4.0 license.

3.2.    If you are unable to obtain permission from the original copyright holder to publish these figures under the CC BY 4.0 license or if the copyright holder’s requirements are incompatible with the CC BY 4.0 license, please either i) remove the figure or ii) supply a replacement figure that complies with the CC BY 4.0 license. Please check copyright information on all replacement figures and update the figure caption with source information. If applicable, please specify in the figure caption text when a figure is similar but not identical to the original image and is therefore for illustrative purposes only.

Reviewers' comments:

Reviewer's Responses to Questions

**Comments to the Author**

1. Is the manuscript technically sound, and do the data support the conclusions?

Reviewer #1: Yes

Reviewer #2: Yes

2. Has the statistical analysis been performed appropriately and rigorously? 

Reviewer #1: Yes

Reviewer #2: Yes

3. Have the authors made all data underlying the findings in their manuscript fully available?

Reviewer #1: Yes

Reviewer #2: Yes

4. Is the manuscript presented in an intelligible fashion and written in standard English?

Reviewer #1: Yes

Reviewer #2: Yes

5. Review Comments to the Author

Reviewer #1: THe authors used a group-based local adaptive approach to enhance the existing deep multiple kernel Learning (DMKL) method toenhance the classification accuracy . Four basic kernels ,

Arc-cosine kernel then small sample sizes and low dimensions UCI datasets and Caltech-256 dataset with large sample sizes and high dimensions have been utilized to evaluate the performance of the GLDMKL approach.

The authors provide several analyses and comparisons therefore i recommend the manuscript to be published in the journal

Reviewer #2: The authors proposed new algorithms called DMKL. In general the paper is well written but very poor formatted. Moreover, the following issues must be addressed:

1) The quality of images must be improved.

2) The quality of charts must be improved.

3) Used bibliography must be changed. Use mainly the papers from the last 3-4 years to show the current state of knowledge.

4) More explanation is needed for what exactly is the novelty of this paper -- it must be more underlined in the abstract and related works.

5) Backgound sections must be merged to related works.

6) Some theoretical analysis of the convergence of your solution is needed.

7) More experiments must be added.

8) More comparisons with the lastest solutions are needed.

9) More statistical analysis should also be added and discussed.

6. PLOS authors have the option to publish the peer review history of their article (what does this mean?). If published, this will include your full peer review and any attached files.

Reviewer #1: **Yes: **dalia yousri

Reviewer #2: No

---

## [Author Response · Author response to Decision Letter 0]

13 Aug 2020

Dear Reviewers，

     Thank you for your review of our manuscript (ID: PONE-D-20-09329), which is titled “Group-based local adaptive deep multiple kernel learning with lp norm”. We appreciate your concerns and suggestions, and have revised our manuscript accordingly.

 The followings are our responses for your comments.

Point1:

Response1: 

 We supplement and refine the experimental data, add corresponding analysis and further explain the ambiguities. All the modifications are marked in Yellow in the paper.

Point2:

Has the statistical analysis been performed appropriately and rigorously?

Response2: 

Thanks to the reviewers for affirming that there are detailed statistical analysis in my paper. In the experiment of UCI datasets, we have discussed comparison of classification performance, the effect of group numbers on classification performance and the effect of layer numbers on classification performance. In the experiment of Caltech-256 datasets, we randomly select five types of data with similar shapes: bowling-ball, car-tire, desk-globe, roulette-wheel and sunflower-101. When these five datasets are respectively used as positive, we have analyzed the classification accuracy under different numbers of layers and groups. Special thanks for your good suggestions.

Point3:

Response3: 

The data in my experiment has been provided as part of the paper. There are no restrictions on publicly sharing data. The url is: https://github.com/fage8/GLDMKL.git.

Point4:

Response4: 

The language usage, spelling and grammar have been completely revised in the paper. All the modifications are marked in Yellow in the paper. For example, lp norm is modified to the lp norm. classical DMKL is modified to the classical DMKL.

Point5:

The authors provide several analyses and comparisons therefore i recommend the manuscript to be published in the journal.

Response5: 

Thank your recommendation about our paper. We make further modifications to our paper. We supplement and refine the experimental data, add corresponding analysis and further explain the ambiguities.

Point6:

 The quality of images must be improved.

Response6: 

Figures 1-5 and Figure 11 have been modified, and some key elements in the figures have been modified and added with color labels. Figures 6-10 have been removed from the paper.

Point7:

The quality of charts must be improved.

Response7: 

All tables in the paper are changed to three-line tables, and their captions and header elements are marked in bold.

Point8:

Used bibliography must be changed. Use mainly the papers from the last 3-4 years to show the current state of knowledge.

Response8: 

Thank you for your excellent suggestions. We add the papers from the last 3-4 years as references. All the modifications are marked in Yellow in the paper.

Point9:

More explanation is needed for what exactly is the novelty of this paper -- it must be more underlined in the abstract and related works.

Response9: 

The novelties of this paper are underlined in the abstract and related works. The novelties of this paper are the followings: our GLDMKL method can divide samples into multiple groups according to the multiple kernel k-means clustering algorithm. The learning process in each well-grouped local space is exactly adaptive deep multiple kernel learning. And our structure is adaptive, so there is no fixed number of layers. The learning model in each group is trained independently, so the number of layers of the learning model maybe different which highlights the flexibility of the model. Our model is more adaptable to data of different dimensions and sizes. All the modifications are marked in Yellow in the paper.

Point10:

Backgound sections must be merged to related works.

Response10: 

Special thanks for your good suggestions. Background section is merged to related works. Some descriptions about DMKL have been integrated and some new references have been added.

Point11:

Some theoretical analysis of the convergence of your solution is needed.

Response11: 

 We add some theoretical analysis of the convergence of our method. In our method, we transform the non-convex problem into a convex function problem through a gate function. In this way, the local minimum must be found by the gradient ascent method, so our method must converge.

Point12:

More experiments must be added.

Response12: 

The latest solutions are added in our paper. The latest method DWS-MKL in 2020 is added to the classification accuracy comparison experiment. Specifically, these added experiments include the comparison experiment of best classification performance and the comparison experiment of classification performance at each layer.

Point13:

More comparisons with the lastest solutions are needed.

Response13: 

The latest method DWS-MKL in 2020 is added to the classification accuracy comparison experiment. Specifically, these comparisons include the comparison of best classification performance and the comparison of classification performance at each layer. In comparison experiment, the classification performance of our GLDMKL method is better than DWS-MKL method on various datasets.

 Point14:

More statistical analysis should also be added and discussed.

Response14: 

We have discussed the effect of group numbers and the effect of layer numbers. Some statistical analysis will be added in futrue work in Validation section. For example, the probabilistic grouping can be added in our model, the influence of the lp norm can be discussed and the stability of model can be discussed.

More reasons and modifications have been noted in the article, all the modifications are marked in Yellow in the paper. Special thanks to you for your good comments.

Sincerely yours,

Shengbing Ren, Fa Liu, Weijia Zhou, Xian Feng and Chaudry Naeem Siddique

---

## [Decision Letter · Decision Letter 1]

19 Aug 2020

Group-based local adaptive deep multiple kernel learning with lp norm

PONE-D-20-09329R1

Dear Dr. Ren,

We’re pleased to inform you that your manuscript has been judged scientifically suitable for publication and will be formally accepted for publication once it meets all outstanding technical requirements.

Kind regards,

Robertas Damasevicius

Academic Editor

PLOS ONE

Additional Editor Comments (optional):

Reviewers' comments:

Reviewer's Responses to Questions

**Comments to the Author**

1. If the authors have adequately addressed your comments raised in a previous round of review and you feel that this manuscript is now acceptable for publication, you may indicate that here to bypass the “Comments to the Author” section, enter your conflict of interest statement in the “Confidential to Editor” section, and submit your "Accept" recommendation.

Reviewer #1: All comments have been addressed

Reviewer #2: All comments have been addressed

2. Is the manuscript technically sound, and do the data support the conclusions?

Reviewer #1: Yes

Reviewer #2: Yes

3. Has the statistical analysis been performed appropriately and rigorously? 

Reviewer #1: Yes

Reviewer #2: Yes

4. Have the authors made all data underlying the findings in their manuscript fully available?

Reviewer #1: Yes

Reviewer #2: Yes

5. Is the manuscript presented in an intelligible fashion and written in standard English?

Reviewer #1: Yes

Reviewer #2: Yes

6. Review Comments to the Author

Reviewer #1: All the comments have been addressed

I dont have another additional comments therefore the manuscript should be accepted in the journal.

Reviewer #2: The paper can be accepted. All comments have been addressed and the quality of the paper is ok for PLOS ONE.

7. PLOS authors have the option to publish the peer review history of their article (what does this mean?). If published, this will include your full peer review and any attached files.

Reviewer #1: No

Reviewer #2: No

---

## [Editor Report · Acceptance letter]

8 Sep 2020

PONE-D-20-09329R1 

Group-based local adaptive deep multiple kernel learning with lp norm 

Dear Dr. Ren:

I'm pleased to inform you that your manuscript has been deemed suitable for publication in PLOS ONE. Congratulations! Your manuscript is now with our production department. 

Kind regards, 

on behalf of

Professor Robertas Damasevicius 

Academic Editor

PLOS ONE